# PhysX-3D: Physical-Grounded 3D Asset Generation

Ziang Cao[1]   Zhaoxi Chen[1]   Liang Pan[2]   Ziwei Liu[1]*
[1]Nanyang Technological University   [2]Shanghai AI Lab
https://physx-3d.github.io/

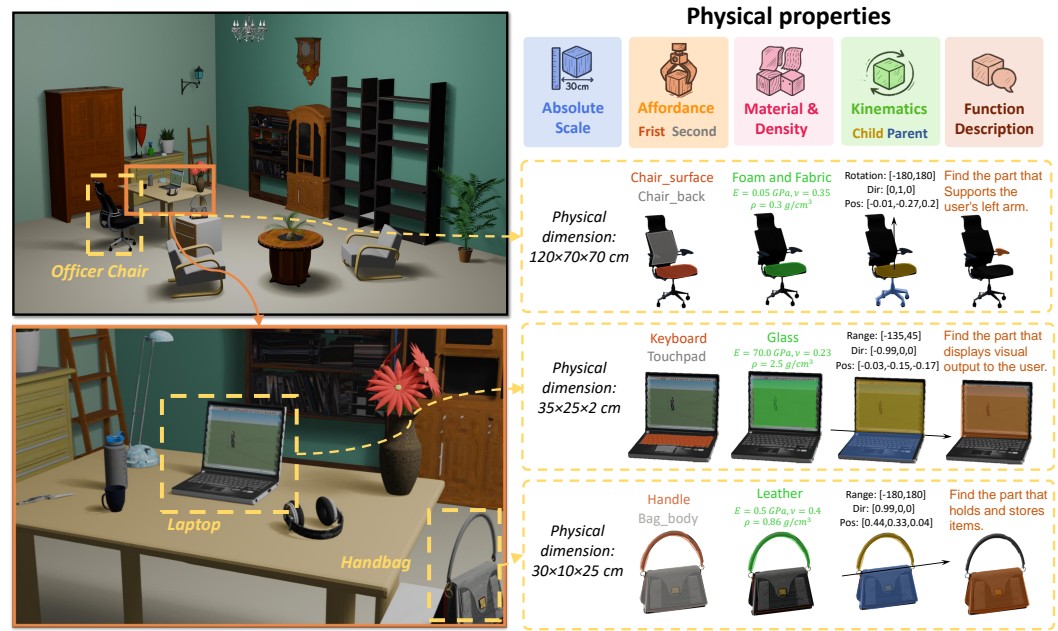

Figure 1: Visualizations of our PhysXNet for phsycial 3D generation. 3D assets in our dataset have fine-grained physical property annotations, including **1) absolute scale**, **2) material**, **3) affordance**, **4) kinematics**, and **5) function descriptions** (basic, functional, and kinematical descriptions).

## Abstract

3D modeling is moving from virtual to physical. Existing 3D generation primarily emphasizes geometries and textures while neglecting physical-grounded modeling. Consequently, despite the rapid development of 3D generative models, the synthesized 3D assets often overlook rich and important physical properties, hampering their real-world application in physical domains like simulation and embodied AI. As an initial attempt to address this challenge, we propose **PhysX**, an end-to-end paradigm for physical-grounded 3D asset generation. **1)** To bridge the critical gap in physics-annotated 3D datasets, we present **PhysXNet** - the first physics-grounded 3D dataset systematically annotated across five foundational dimensions: **absolute scale**, **material**, **affordance**, **kinematics**, and **function description**. In particular, we devise a scalable human-in-the-loop annotation pipeline based on vision-language models, which enables efficient creation of physics-first assets from raw 3D assets. **2)** Furthermore, we propose **PhysXGen**,

---

*Corresponding author, ziwei.liu@ntu.edu.sg

39th Conference on Neural Information Processing Systems (NeurIPS 2025).

a feed-forward framework for physics-grounded image-to-3D asset generation, injecting physical knowledge into the pre-trained 3D structural space. Specifically, PhysXGen employs a dual-branch architecture to explicitly model the latent correlations between 3D structures and physical properties, thereby producing 3D assets with plausible physical predictions while preserving the native geometry quality. Extensive experiments validate the superior performance and promising generalization capability of our framework. All the code, data, and models will be released to facilitate future research in generative physical AI.

# 1 Introduction

The creation of diverse and high-quality 3D assets has gained significant prominence in recent years, driven by their expanding applications across gaming, robotics, and embodied simulators. Substantial research efforts have been focused on appearance and geometry only, from high-quality 3D datasets [1, 2, 3, 4], efficient 3D representations, to generative modeling. However, most of them predominantly emphasize structural characteristics while overlooking physical properties inherent to real-world objects. Given the rising demand for physical modeling, understanding, and reasoning in 3D space, we argue that a comprehensive suite for physics-grounded 3D objects is important, from upstream data annotations pipeline to downstream generative modeling.

Beyond purely structural attributes like geometry and appearance, real-world objects intrinsically possess rich physical and semantic characteristics comprising: **1) absolute scale**, **2) material**, **3) affordance**, **4) kinematics**, and **5) function descriptions**. By integrating these fundamental properties with classical physical principles, we can derive critical dynamic metrics, including gravitational effects, frictional forces, contact region, motion trajectories, and interaction. However, existing datasets/annotation pipelines only offer partial solutions towards physically grounded knowledge in 3D objects that cover the entire spectrum. Recent efforts to support articulated object applications have yielded datasets like PartNet-Mobility [5], which provides 2.7K human-annotated articulated 3D models. Yet, this collection still lacks essential physical descriptors - including dimensional specifications, material composition, and functional affordances - that are crucial for physically accurate simulations and robotics applications.

To bridge this representational gap, we propose **PhysXNet** – the first comprehensive physical 3D dataset containing over 26K richly annotated 3D objects, as illustrated in Figure 1. Except for the object-level annotation, *i.e.*, 1), we annotate 2) and 5) for each part. Besides, for 3), we provide the affordance rank for all parts, while we annotate the 4) detailed parameters of kinematic constraints, including motion range, motion direction, child parts, and parent parts. Besides, we introduce an extended version, **PhysXNet-XL**, featuring over 6 million procedurally generated and annotated 3D objects.

Most importantly, PhysXNet is built with an efficient, robust, and scalable labeling pipeline. We introduce a human-in-the-loop annotation pipeline to annotate the properties for the existing object-level 3D dataset, *i.e.*, PartNet [6]. The pipeline proceeds in three stages: 1) target visual isolation, in which we render each component via alpha compositing to get the best visual prompts with minimized visual interference. 2) automatic VLM labeling, where a large vision-language model (VLM) to annotate most of the properties; and 3) expert refinement, combining systematic spot-checks with focused human annotation of complex kinematic behaviors. To the best of our knowledge, PhysXNet is the first 3D dataset with abundant properties for each part.

To bridge the modeling gap of physical-grounded 3D assets, we further introduce **PhysXGen**, a feedforward model for physical 3D generation. Given the fact that physical properties are spatially related to geometry and appearance, we repurpose pretrained 3D generative priors to generate physical 3D assets, enabling efficient training with reasonable generalizability. Specifically, PhysXGen leverages a dual-branch architecture to jointly model the latent correlations between 3D geometric structures and physical properties, which is naturally compatible with existing 3D native generative priors. Moreover, this formulation makes the best use of pretrained latent space, leading to plausible physical predictions while keeping the decent geometry quality from the pretrained model. Comprehensive experiments prove the promising performance of PhysXGen. We hope our work reveals new observations, challenges, and potential directions for future research in embodied AI and robotics.

To summarize, our main contributions are:

Table 1: Comparison of related datasets which can support research in physical 3D generation. While the ABO dataset [7] contains material metadata and keywords, its object-level annotation granularity constrains part-aware applications like robotic manipulation or physical simulation. In contrast, PhysXNet provides part-level annotations.

| Dataset | # Objs | Part anno | Physical Dim | Material | Affordance | Kinematic | Description | Year |
|---|---|---|---|---|---|---|---|---|
| ShapeNet [1] | 51K | ✗ | ✗ | ✗ | ✗ | ✗ | ✗ | 2015 |
| PartNet [6] | 26K | ✓ | ✗ | ✗ | ✗ | ✗ | ✗ | 2019 |
| PartNet-Mobility [5] | 2.7K | ✓ | ✗ | ✗ | ✗ | ✓ | ✗ | 2020 |
| GAPartNet [8] | 1.1K | ✓ | ✗ | ✗ | ✗ | ✓ | ✗ | 2022 |
| ABO [7] | 7.9K | ✗ | ✓ | Obj-level | ✗ | ✗ | Obj-level | 2022 |
| OmniObject3D [4] | 6K | ✗ | ✗ | ✗ | ✗ | ✗ | ✗ | 2023 |
| Objaverse [2] | 818K | ✗ | ✗ | ✗ | ✗ | ✗ | ✗ | 2023 |
| **PhysXNet (ours)** | 26K | ✓ | ✓ | Part-level | ✓ | ✓ | Part-level | 2025 |
| **PhysXNet-XL (ours)** | 6M | ✓ | ✓ | Part-level | ✓ | ✓ | Part-level | 2025 |

- We pioneer the first end-to-end paradigm for physical-grounded 3D asset generation, advancing the research frontier in physical-grounded content creation and unlocking new possibilities for downstream applications in simulation.
- We build the first physical-grounded 3D dataset, **PhysXNet**, and propose a human-in-the-loop annotation pipeline to convert existing geometry-focused datasets into fine-grained physics-annotated 3D datasets efficiently and robustly. In addition, we present an extended version, **PhysXNet-XL**, which includes over 6 million annotated 3D objects generated through procedural methods.
- We design a dual-branch feed-forward framework, **PhysXGen**. It can model the latent interdependencies between structural and physical features to achieve plausible physical predictions while maintaining the native geometry quality.

## 2 Related Work

### 2.1 3D Datasets and Benchmarks

Due to the time-consuming and expensive in realistic data collection, current large-scale 3D datasets prefer to collect data online [1, 2, 3]. According to the type of 3D data, existing 3D datasets can be divided into synthetic and real-world datasets. To facilitate the development of 3D vision, ShapeNet [1] collects 51,300 CAD models. Building upon it, the PartNet dataset [6] introduces an annotation framework that provides part annotations at significantly finer granularity levels. Furthermore, PartNet-Mobility [5] annotates the kinematic constraints and provides 2.7K articulated 3D objects for 3D vision, especially for embodied AI and robotics. ABO [7] is a high-quality datasets with around 7.9K CAD models with fine-grained geometric and textures. Compared with prior work, it includes the physical dimension, material, and keywords. However, the material information and descriptions focus on object-level, limiting the part-aware applications. Recently, Objaverse [2] has alleviated the scarcity of 3D data. It collects and filters over 800K 3D data. To bridge the gap between synthetic and real data, Omniobject3D [4] provides over 6k high-quality 3D scans. A detailed comparison is shown in Table 1.

Despite significant advances in 3D data acquisition, prevailing 3D datasets primarily emphasize geometry and appearance fidelity or narrowly defined physical attributes, creating a critical bottleneck for developing physics-aware 3D vision models and their real-world applications. To bridge this foundational gap, we present PhysXNet – a 3D dataset with comprehensive physical properties encompassing physical dimension, part-level material, affordance rank, kinematic parameters, and part-level description. Furthermore, we extend our dataset with **PhysXNet-XL**, comprising more than 6 million annotated 3D objects created via procedural generation.

### 2.2 3D Generative Models

As one of the most representative optimization-based method in 3D generation, DreamFusion [9] proposed the SDS loss function. By utilizing the prior knowledge of the 2D diffusion model, it achieves impressive generative performance. Despite various works, optimization-based methods still suffer from the multi-face Janus problem and low optimization efficiency. Recently, benefiting from

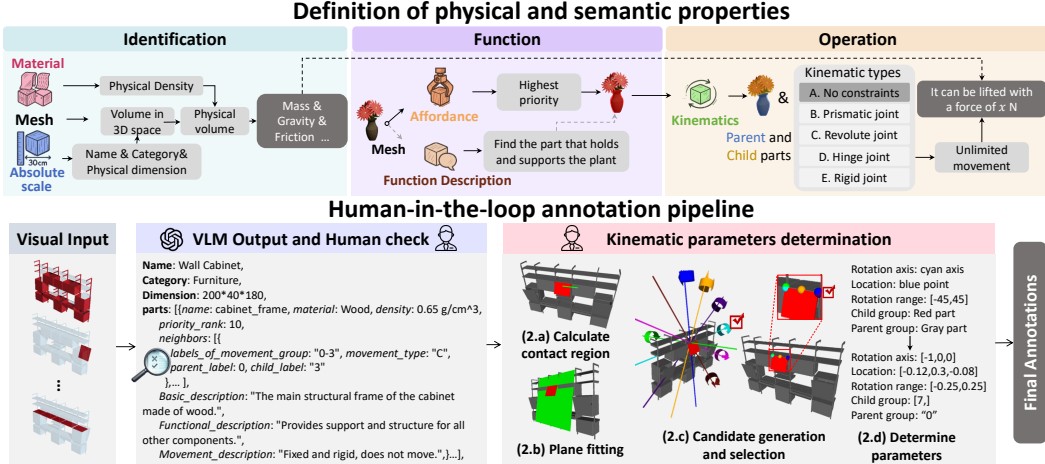

Figure 2: **Top: Definition of properties in PhysXNet .** By defining and annotating properties across three categories, common physical quantities can be systematically calculated to enable physical simulations. **Bottom: Overview of our human-in-the-loop annotation pipeline.** We utilize GPT-4o to gather foundational raw data, which is subsequently verified through human oversight. The kinematic parameters are then rigorously determined and finalized through human review.

its impressive efficiency and robustness, feed-foward models [10, 11, 12, 13, 14, 15, 16, 17] have gained more and more attention. However, those methods still focus on geometry and appearance quality, neglecting the physical properties of 3D assets.

## 2.3 Articulated and Physical 3D Object Modeling

Articulated object modeling mainly consists of tasks like perception, reconstruction, and generation. Some works try to estimate articulation pose [18] and identify articulation parts [19], while others [20] focus on learn joint parameters from images. In the reconstruction field, existing works try to reconstruct articulated models from RGB [21], RGBD [22], and point cloud [23]. Recently, some methods have tried to generate articulated 3D assets by utilizing a vision-language model [24, 25] or adopting an optimization-based framework [26]. To bridge the critical gap between existing methods with real applications, many works aim to incorporate the physical properties into 3D modeling. Some works try to learn material parameters from videos [27] or images [28], while other methods aim to introduce physical guidance via simulation [29, 30] or physical principles [31].

In contrast to fragmented paradigms in physical 3D modeling, this work introduces PhysXGen – a unified physics-integrated generative framework capable of learning cross-property consistency to generate 3D assets with all necessary physical properties. By exploiting the relationship between physical and structural features, our method achieves promising performance in physical 3D generation.

## 3 PhysXNet Dataset

In this section, we will introduce physical properties and the human-in-the-loop annotation pipeline. Besides, we will report the statistics and distribution of PhysXNetand PhysXNet-XL.

### 3.1 Definition of Physical Properties

As shown in Figure 2, we systematically categorize object properties into three progressive stages: a) Identification - determining the basic nature of the object; b) Function - understanding its potential applications; and c) Operation - detailed usage methodologies. To streamline the annotation process, we posit that the internal composition of a component is homogeneous, exhibiting uniform property

invariance throughout its structure. For stages a), we set **absolute scaling** and **material** (material name, Young's modulus, Poisson's ratio, and density). Besides, for b), we establish functional **affordance** analysis and **function descriptions** (basic, functional, and kinematic descriptions). Finally, we use **kinematic** parameter quantification to represent c). Specifically, we grade the priority of being touched on all available parts to obtain the affordance score for all parts from 1 to 10. We set five possible kinematic types: A. No movement constraints (like water in a bottle), B. Prismatic joints (like a drawer), C. Revolute joints (like a laptop), D. Hinge joint (like a hose in a shower system), or E. Rigid joint and a combined kinematic type: CB. Revolute and Prismatic joints (like a lid of a bottle). Except for A and E, we will annotate the parent, child parts, and detailed kinematic parameters (such as rotation direction, rotation range, and so on). Note that, due to the challenges in precisely quantifying the absolute physical movement range of B, we use the movement range within the 3D coordinate system. Besides, to avoid the unnecessary and meaningless annotation of over-fine-grained parts in PartNet, we merge the tiny parts whose vertices and area are smaller than a pre-defined threshold with their neighboring parts. We manually refine the results of the merging process to ensure that the merged outputs are reasonable and consistent.

### 3.2 Human-in-the-loop Annotation Pipeline

Following the establishment of target annotation specifications, we implement a systematic and streamlined semi-automated annotation framework, structured into two distinct operational phases (see Figure 2): 1) Preliminary Data Acquisition and 2) Kinematic Parameter Determination. Specifically, we utilize GPT-4o to obtain the basic information. Besides, to ensure the quality of raw data, a human candidate will check and refine the output of the vision-language model (VLM).

For the second phrase, we split it into four subtasks: (2.a) calculate contact region, (2.b) plane fitting, (2.c) candidate generation and selection, and (2.d) kinematic parameters. Note that (2.c) and (2.d) are accomplished by human candidate. For all constraint movable parts (kinematic type is not A or E), we will calculate the contact region with the neighboring parts. We first extract point cloud data from the child-parent mesh pair, formally designated as $P_c$ and $P_p$, respectively. The workflow subsequently calculates Euclidean distance between points in $P_c$ and $P_p$, followed by spatial filtration that eliminates point pairs failing to meet a predetermined distance threshold. Subsequently, we employ a plane-fitting algorithm. We sample several axes uniformly on the fitted plane as candidates. Note that for kinematic type C, we additionally need to determine the location of the rotation axis. Therefore, we will perform a k-means algorithm in the contact region for type C to generate several candidates. After selecting the candidate location, we can finalize the kinematic parameters.

### 3.3 Statistics and Distribution of PhysXNet

Comprises over 26K physical 3D objects, the part number of objects in PhysXNet exhibits a long-tailed distribution illustrated in Figure 3, where each object contains an average of around 5 constituent parts. Besides, we document the length-width-height distributions of objects in (b). Given that PhysXNet encompasses objects spanning from relatively small-scale indoor entities to large-scale outdoor structures, the physical dimension exhibits significant variation among objects. For kinematic types and material in PhysXNet, we show detailed proportional composition. Note the density in our PhysXNetadheres to the metric standardization framework, *i.e.*, $g/cm^3$. Furthermore, Figure 3 (d) shows the frequency of the popular object tags, including the name and category. Finally, we also report the component category in our procedurally generated 3D objects, including a) intra-category combination: cabinet, bottle, faucet, chair, oven, shower, knife, table, and laptop; b) cross-category combination: drawer and door. More details about PhysXNet-XL are released in the appendix.

## 4 PhysXGen Framework

As mentioned above, physical 3D generation is still a challenging and promising task. Most prior works only focus on a single or specific physical property. In this section, we aim to build a unified generative framework to generate physical 3D assets directly. While our PhysXNet dataset contains 26K assets, this scale remains insufficient for training SOTA generative architectures from scratch. Therefore, we leverage a model pre-trained on massive geometry-only 3D scans and fine-tune it to adapt to physical 3D generation. Building upon the well-established 3D representation space of it, we present PhysXGen, a novel yet straightforward framework that combines physical properties

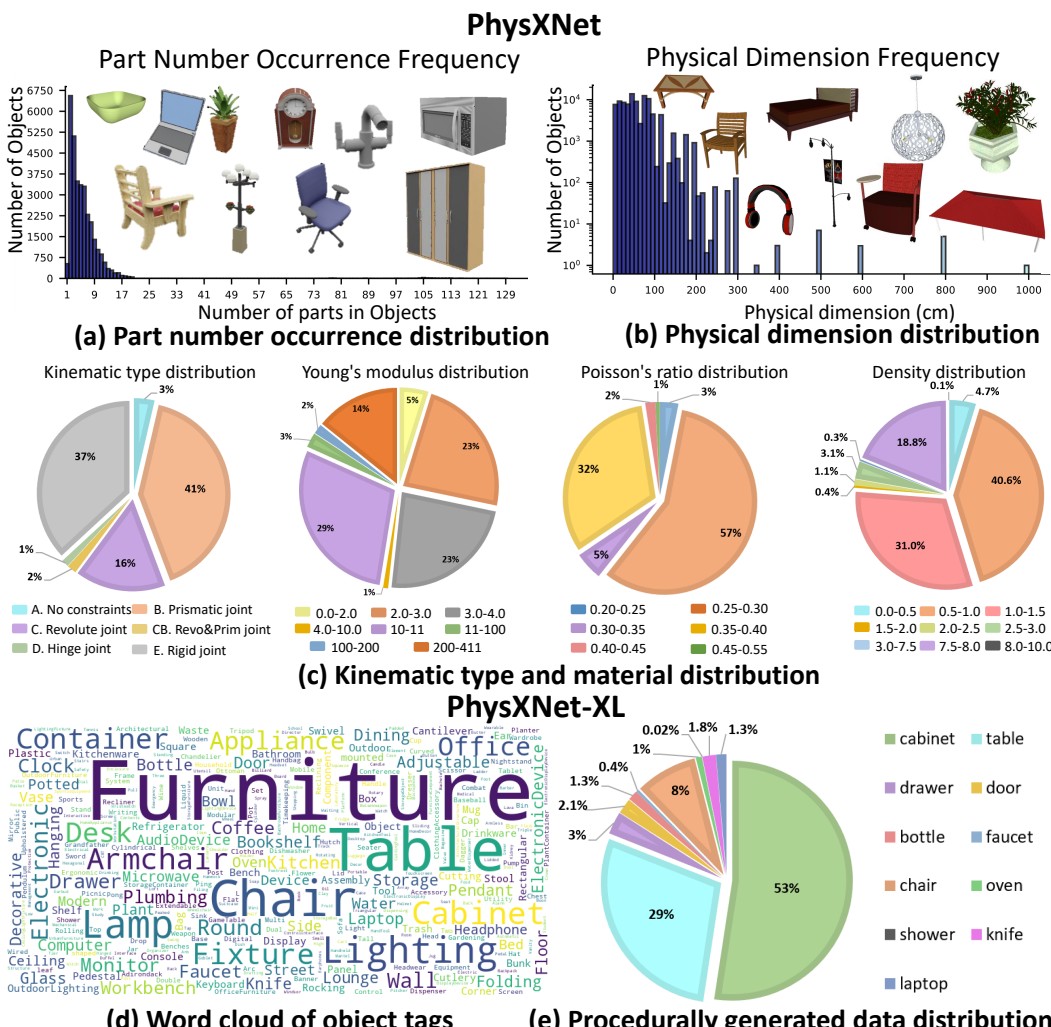

Figure 3: **Statistics and distribution of PhysXNet and PhysXNet-XL**. (a) Distribution histogram of part number in PhysXNet. (b) Dimensional distribution analysis in PhysXNet, showing physical measurements (length/width/height) frequency. (c) Proportional composition of kinematic states and material, including density, Young's modulus, and Poisson's ratio distribution in PhysXNet, visualized through sectoral ratios. (d) Tag frequency statistics for prevalent object labels in PhysXNet-XL. (e) Component-Category distribution of procedurally generated 3D objects in PhysXNet-XL.

with geometry and appearance shown in Figure 4. Our approach achieves this dual objective by simultaneously integrating fundamental physical properties into the generation process while optimizing the structural branch through targeted fine-tuning. This joint optimization enables the production of physically consistent 3D assets that maintain impressive geometry and appearance fidelity.

## 4.1 Physical 3D VAE Encoding and Decoding

In this subsection, we take the textured mesh output as an example. To reduce the influence caused by the domain gap between geometric and physical latent space, we build a similar physical VAE for property encoding, following [10]. Besides, considering the interdependencies among physical properties, we encode them into a unified latent space. We adopt 4 physical properties: physical scaling (converted by physical dimension) $P_{dim} \in \mathbb{R}^{N \times 1}$, affordance priority $P_{aff} \in \mathbb{R}^{N \times 1}$, density $P_{\rho} \in \mathbb{R}^{N \times 1}$, and kinematic parameters $P_{mov} \in \mathbb{R}^{N \times 11}$ (including child $\mathbb{R}^{N \times 1}$ and parent group

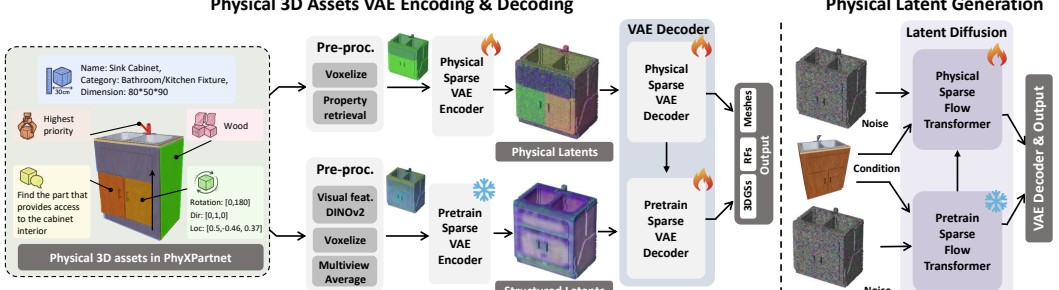

Figure 4: **The architecture of PhysXGen framework.** PhysXGen features a two-stage architecture comprising: a physical 3D VAE framework for latent space learning, and a physics-aware generative process for structured latent. The former focuses on establishing a compressed yet information-rich latent representation that encodes physical properties, while the latter specializes in generating physical latents.

index $\mathbb{R}^{N \times 1}$, movement direction $\mathbb{R}^{N \times 3}$, movement location $\mathbb{R}^{N \times 3}$, movement range $\mathbb{R}^{N \times 2}$, and kinematic type $\mathbb{R}^{N \times 1}$), where N is the number of voxel. The physical properties ($P_{phy} \in \mathbb{R}^{N \times 14}$) can be obtained by channel-wise concatenation. For the function descriptions, we adopt the CLIP model [32] to obtain the text embedding. Similarly, the description features ($P_{sem} \in \mathbb{R}^{N \times 768 \times 3}$) are formed by concatenating the basic, functional, and kinematic description embeddings. Besides, the structural branch adopts the DINOv2 to extract features. Therefore, the dimensions of structural feature is $P_{aes} \in \mathbb{R}^{N \times 1024}$. For clarification, we denote the pretrain VAE encoder and decoder as $\mathcal{E}_{aes}$ and $\mathcal{D}_{aes}$ while the physical VAE encoder and decoder as $\mathcal{E}_{phy}$ and $\mathcal{D}_{phy}$. The physical latent $P_{plat} \in \mathcal{R}^{N \times 8}$ and structured latent $P_{slat} \in \mathcal{R}^{N \times 8}$ can be formulated as follows:

$$P_{plat} = \mathcal{E}_{phy}(P_{phy}, P_{sem}), \ P_{slat} = \mathcal{E}_{aes}(P_{aes}) \ . \tag{1}$$

To study the effects of physical properties on geometry and appearance quality, we introduce a branch from $\mathcal{D}_{phy}$ to $\mathcal{D}_{aes}$ via a residual connection. We will analyze the performance of the independent and dependent VAE decoder in the experiments. After decoding the structured and physical latents, we can implement a loss function $\mathcal{L}$ as follows:

$$\mathcal{L}_{vae} = \mathcal{L}_{aes}^{color} + \mathcal{L}_{aes}^{geometry} + \mathcal{L}_{phy} + \mathcal{L}_{sem} + \mathcal{L}_{kl} + \mathcal{L}_{reg} \ , \tag{2}$$

where $\mathcal{L}_{aes}^{color}$ and $\mathcal{L}_{aes}^{geometry}$ represent the color loss (including L2loss, lpip loss) and geometry loss (including mask, normal, and depth loss). For $\mathcal{L}_{phy}$ and $\mathcal{L}_{sem}$, we normalize the groundtruth respectively and adopt a L2 loss. $\mathcal{L}_{kl}$ aims to constrain the distribution of $P_{plat}$ while $\mathcal{L}_{reg}$ can reduce the unnecessary structures of textured mesh.

## 4.2 Physical Latent Generation

Following the acquisition of the compressed physical latent representation, we construct a transformer-architecture diffusion model to jointly generate physical and structural attributes. To effectively leverage the inherent correlations between physical properties and structural features while maintaining compatibility with pre-trained components, we implement a dual-branch architecture that integrates structural guidance through residual connections. Specifically, the additional branch from the structural module is fused with the primary physical generation module via learnable skip-connection layers, enabling cross-domain feature interaction. Comprehensive ablation studies quantitatively validate the design rationale through systematic component comparisons. Following [10], we adopt the Conditional Flow Matching (CFM) as the objective of optimization. Therefore, the loss of the geometric branch is formulated:

$$\mathcal{L}_{aes} = \mathbb{E}_{t,x_0,\epsilon} ||f(x,t) - (\epsilon - x_0)||_2^2 \ , \tag{3}$$

where $\epsilon$ and $t$ represent the noise and timestep while $x_0$ is sampled from $P_{slat}$. Adopting a similar objective for the physical branch, the final loss of the latent diffusion model can be calculated as: $\mathcal{L}_{diff} = \mathcal{L}_{aes} + \mathcal{L}_{phy}$.

Table 2: Quantitative comparison of different methods on the test sets of our PhysXNet. There are two types of evaluations: structural and physical property evaluations. PhysPre represents a separate physical property predictor after TRELLIS.

| Methods | Geometry | | | Absolute scale ↓ | Material ↑ | Affordance ↑ | Kinematic parameters | | Description ↑ |
|---------|------|------|---------|------------------|-----------|--------------|---------|---------|--------------|
| | PSNR ↑ | CD ↓ | F-Score ↑ | | | | COV ↑ | MMD ↓ | |
| TRELLIS [10] | 24.31 | 13.2 | 76.9 | – | – | – | – | – | – |
| TRELLIS + PhysPre | 24.31 | 13.2 | 76.9 | 13.21 | 8.63 | 7.23 | 0.24 | 0.12 | 6.55 |
| PhysXGen | **24.53** | **12.7** | **77.3** | **7.24** | **13.01** | **11.30** | **0.33** | **0.08** | **10.11** |

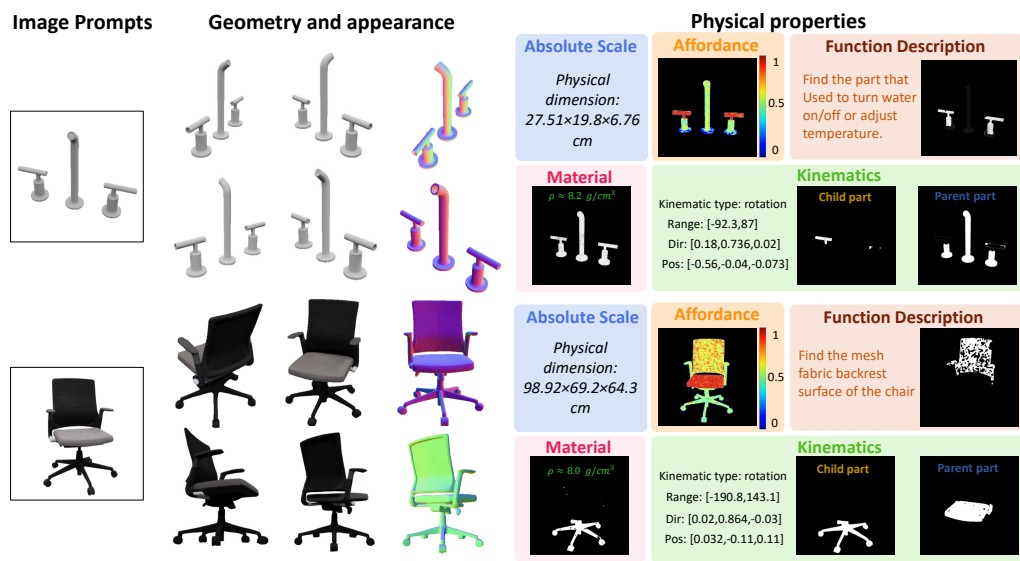

Figure 5: **Visualization of the generated results.** Given a single image as the prompt, our PhysXGen can generate the physical-grounded 3D assets.

## 5 Experiments

### 5.1 Implementation details

In our experiments, we partition PhysXNet dataset into 24K training samples, 1K validation samples, and 1K test cases. By analyzing the performance on the test cases, we can evaluate the generalizability of our method. During the VAE and diffusion model training, we adopt AdamW with an initial learning rate of $1 \times 10^{-4}$ to optimize the models. The inherent correlation between geometric configuration and physical properties in our methodology creates a critical dependency where the structural fidelity of the 3D representation will affect the final generative performance. In this paper, we repurpose the geometry- and appearance-rich structural space of TRELLIS [10] for our task. Our PhysXGen is trained on 8 NVIDIA A100 GPUS. More details about the architecture are released in the supplementary.

### 5.2 Evaluation Metrics

**Physical properties evaluation.** Our framework establishes a multi-property feature space encompassing five core attributes: **absolute scale**, **material**, **affordance**, **kinematics**, and **function descriptors**. Note that the **kinematics** attribute manifests as dual configuration parameters: 1) structural grouping (parent-child part hierarchies) and 2) kinematic parameters. Specifically, we evaluate absolute scale using Euclidean distance, density and affordance images via Peak Signal-to-Noise Ratio (PSNR), kinematics with instantiation distance [33], and functional description through PSNR on cosine similarity score maps.

**Geometry evaluation.** For appearance evaluation, we sample 30 random views from a unit sphere to calculate the mean PSNR. Besides, to evaluate the quality of geometry, we calculate the standard shape metrics of Chamfer Distance (CD) ($\times 10^{-3}$) and F-score (FS) ($\times 10^{-2}$) with thresholds of 0.05.

Table 3: Ablation studies about the physical 3D VAE and diffusion model. Dep-VAE and Dep-Diff represent the model that utilizes the interdependencies between structural and physical information. Thus, Trellis+PhysPre and PhysXGen are corresponding to the first and last lines.

| Dep-VAE | Dep-Diff | Geometry | | | Absolute scale ↓ | Material ↑ | Affordance ↑ | Kinematic parameters | | Description ↑ |
|---|---|---|---|---|---|---|---|---|---|---|
| | | PSNR ↑ | CD ↓ | F-Score ↑ | | | | COV ↑ | MMD ↓ | |
| ✗ | ✗ | 24.31 | 13.2 | 76.9 | 13.21 | 8.63 | 7.23 | 0.24 | 0.12 | 6.55 |
| ✗ | ✓ | 24.31 | 13.2 | 76.9 | 12.01 | 10.69 | 8.95 | 0.26 | 0.11 | 7.71 |
| ✓ | ✗ | 24.32 | 12.9 | 77.0 | 10.57 | 9.86 | 9.32 | 0.28 | 0.11 | 7.54 |
| ✓ | ✓ | 24.53 | 12.7 | 77.3 | 7.24 | 13.01 | 11.30 | 0.33 | 0.08 | 10.11 |

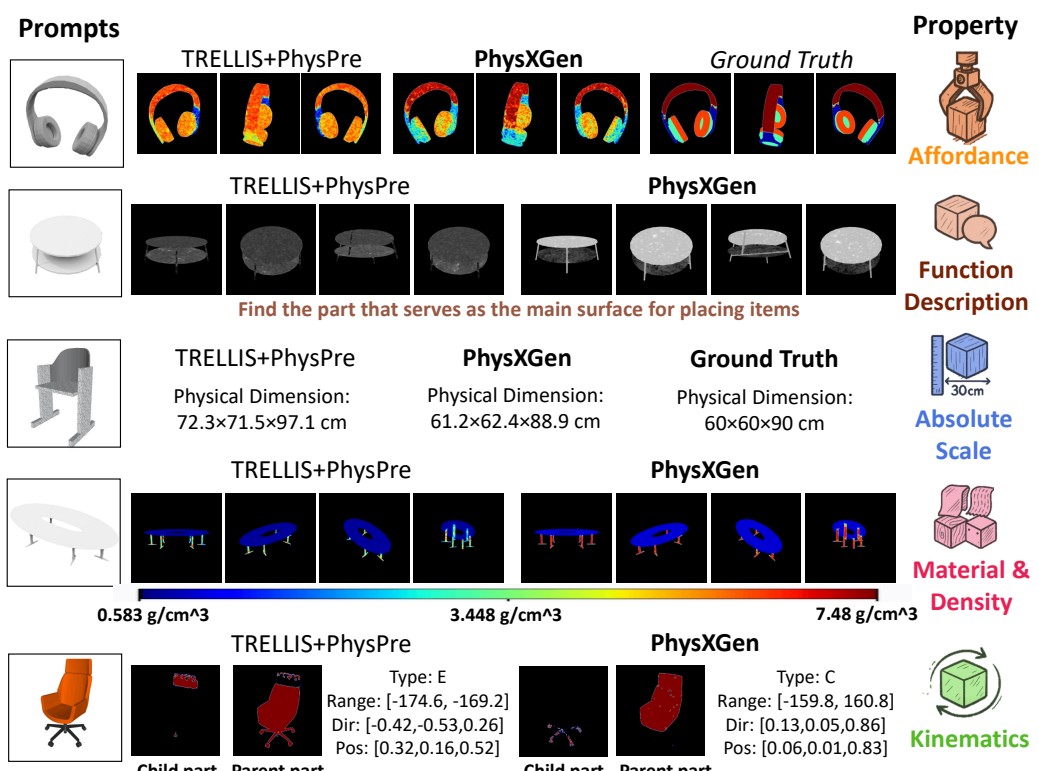

Figure 6: **Qualitative comparison of different methods.** Compared with our baseline, PhysXGen achieves significant improvements, clearly demonstrating its strong performance in physics-grounded 3D generation.

## 5.3 Quantitative Results

As shown in Table 2, we implement the quantitative evaluations on two types of metrics: 1) geometry and appearance evaluation; and 2) physical properties evaluation. Note that TRELLIS+PhysPre is our baseline that adopts the independent structure to predict the properties. Compared with the separate physical property predictor, our PhysXNet utilizes the correlation between physical and pre-defined 3D structural space, achieving significant improvement in physical property generation while enhancing the aesthetic quality.

**Ablation studies.** The core design of our framework is to integrate both geometry and physics in 3D modeling. Therefore, we conduct ablation studies to validate its effectiveness (reported in Table 3). By introducing geometry and appearance features in the diffusion model, the generative model can gain improvement in physics generation compared with the independent models, PhysPre. Additionally, the correlation between geometry and physics in VAE can enhance the geometry of generated assets. Finally, relying on the dual-architecture and joint training, our PhysXGen obtains impressive performance in all physical property generation.

## 5.4 Qualitative Results

Figure 5 showcases the physical-grounded 3D assets generated by our PhysXGen. By learning the interdependencies between physical and structural space, PhysXGen achieves impressive performance in generating physical properties. Besides, we perform qualitative comparisons with our baseline shown in Figure 6. As we mentioned above, for **absolute scaling**, we use the Euclidean distance while we adopt PSNR to evaluate the **material**  maps, **affordance** maps, **function description** similarity score maps. By utilizing the interdependencies between physical properties and structural information, especially geometry, our PhysXNet obtains higher overall scores. Furthermore, our PhysXGen can distinguish the properties of different parts and achieve more stable and robust performance in physical property generation of neighboring structures, especially in **function description**, **material** , and **affordance**. More experimental results are shown in the supplementary.

## 6 Conclusion

In this paper, to fill the gap between existing synthesized 3D assets and real-world applications, we propose an end-to-end generative paradigm for physical-grounded 3D asset generation, including the first physical-grounded 3D dataset and the novel physical property generator. Specifically, we develop a human-in-the-loop annotation pipeline that transforms current 3D repositories into physics-enabled datasets. Meanwhile, the novel end-to-end generative framework, PhysXGen, can integrate physical priors into structural-focused architectures to achieve robust generation performance. Through comprehensive experiments on PhysXNet, we reveal the fundamental challenges and direction in physical 3D generation. We believe that our dataset will attract research attention from different communities, including but not limited to embedded AI, robotics, and 3D vision.

**Limitations and Future works.** Despite impressive performance, our method exhibits limitations in learning fine-grained properties and suffers from artifacts. In our future work, we will try to handle it. Besides, we will include more 3D data from synthetic to real to improve the diversity of our dataset and integrate additional physical properties and kinematic types to better simulate material behavior and movement.

## Acknowledgment

This study is supported by the National Key R&D Program of China (2022ZD0160201), and Shanghai Artificial Intelligence Laboratory. This study is also supported by the Ministry of Education, Singapore, under its MOE AcRF Tier 2 (MOET2EP20221-0012, MOE-T2EP20223-0002). This research is also supported by cash and in-kind funding from NTU S-Lab and industry partner(s).

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
