# OpenReview forum: "PhysX-3D: Physical-Grounded 3D Asset Generation"
_NeurIPS.cc/2025/Conference — NeurIPS 2025 spotlight_

### Official Review · Reviewer_47JR · 2025-06-13

**Clarity:** 2
**Significance:** 4
**Originality:** 3
**Rating:** 5
**Confidence:** 4

**Summary:**

The paper presents a dataset of physical properties of 3D objects, building up on PartNet, including material, size, affordance, kinematics and functionality. The data was generated by prompting an LLM under human supervision. The paper further develops an approach to generate the physical properties within a 3D generation pipeline.

**Questions:**

See weaknesses above, summarized here:
1) How to the labels compare to the ones from prior datasets, e.g. PartnetMobility?
2) What parts of the annotation pipeline, especially the kinematics, is done by humans?
3) How is the MAE between rendered images used to evaluate the property generation?

**Ethical Concerns:**

["NO or VERY MINOR ethics concerns only"]

**Final Justification:**

By doing additional validation studies and integrating further metrics, the authors have fully addressed my concerns.

**Limitations:**

yes

**Quality:**

3

**Strengths And Weaknesses:**

Strengths:
The dataset is very valuable for research on 3D generation, as generating 3D objects for simulations or other applications requires such physical properties. The generative approach is compatible with existing 3D generation methods (e.g. Trellis) and the results demonstrate that the proposed joint approach improves over a separate physical property predictor.

Major weaknesses:
1) The quality of the dataset is not validated in the paper. It depends a lot on the rigor of the human annotator (of 26k objects) whether the annotations are correct, since it is unlikely that GPT-4 has sufficient physical understanding to label these properties on a part level. How much correction was necessary? To verify the validity of the annotations, one could compare selected properties with the annotations from other datasets. For example, the labeled kinematics could be compared to the ones in the PartNetMobility dataset.
2) The methodology is not explained well. Section 3.2 is very brief and not clear to me, e.g. how can k-means determine the location of the rotation axis? How is the type of kinematics (A-E) determined? What tasks in part 2 (kinematic parameter estimation) are done by the human annotator?
3) The metric used (MAE of rendered images) is very unconventional. How are the properties encoded in these images? Why not use suitable per-property metrics from prior work? E.g., Euclidean distance for the absolute scale, text embedding cosine similarity for material and functionality, and suitable measures for kinematics such as instantiation distance [1]?

Minor weaknesses:
1) The dataset is presented as if it was a new 3D dataset, but the objects are entirely taken from PartNet. It should be clarified that this dataset is providing annotations for an existing dataset.
2) Regarding “we merge the tiny parts whose vertices and area are smaller than a pre-defined threshold”: This sounds a bit dangerous with respect to kinematics. What if these merged parts have a non-rigid relationship?

---

> ### Author Rebuttal · Authors · 2025-07-29
>
> We appreciate you for your encouraging comments. We have carefully addressed all your comments one-by-one as follows:
>
> > **Weakness 1**: Lack of a quality comparison for PhysXNet.
>
> **Answer 1**: We sincerely thank you for the detailed comments. Indeed, the quality of the initial GPT-generated annotations is closely correlated with the complexity of the physical property. For relatively straightforward properties such as physical dimensions, material (density), and functional descriptions, the success rate exceeds 90%. For more challenging attributes like affordance, the success rate drops to around 60–70%. For the most difficult property—kinematic information—the total success rate is close to zero. However, it can provide some useful information such as kinematic group information and neighboring parts. Based on it, kinematic-related annotations are manually annoated. To validate the accuracy of the annotations, we conducted a user study comparing PhysXNet with PartNet-Mobility. Participants were asked to evaluate each annotation pair based on three options: (1) PhysXNet is better, (2) PartNet-Mobility is better, or (3) both are equally reasonable. Based on the collected responses, we computed the normalized scores, which demonstrate the high quality of our dataset.
>
>
> | Datasets | Score (Kinematic) |
> |-------|------|
> | Partnet-mobility | 0.502  |
> | PhysXNet   | 0.498 |
>
>
> > **Weakness 2**: Detailed questions about our annotation pipeline.
>
>
> **Answer 2**: Thank you for the thoughtful question. For axis selection, we first use k-means clustering to generate multiple candidate points, from which the final rotation axis is manually selected. The annotations for kinematic A–E are produced via a human-in-the-loop pipeline that combines GPT-generated predictions with manual verification. As for the kinematic properties, all components require manual refinement or are fully annotated (kinematic parameters: location of axis, direction of axis, and kinematic range) by human experts to ensure correctness and consistency. We will include an additional explanation of our annotation pipeline in the new version.
>
> > **Weakness 3**: Unconventional metrics in physical dimension evaluations.
>
> **Answer 3**: We sincerely thank you for this valuable comments. In response, we have revised some evaluation metrics in the paper to better align with different physical aspects. Specifically, we now evaluate absolute scale using Euclidean distance, density and affordance images via PSNR, kinematics with instantiation distance, and functional description through PSNR on cosine similarity score maps. We believe these metrics provide a more comprehensive assessment of the physical properties.
>
> | Methods             | Geometry        |                 |                 | Absolute scale ↓ | Material ↑ | Affordance ↑ | Kinematic parameters |                 | Description ↑ |
> | :------------------ | :-------------  | :-------------  | :-------------  | :--------------- | :--------- | :----------- | :------------------- | :-------------- | :------------ |
> |                     | **PSNR ↑**      | **CD ↓**        | **F-Score ↑**   |                  |            |              | **COV ↑**            | **MMD ↓**       |               |
> | **TRELLIS**    | 24.31           | 13.2            | 76.9            | –                | –          | –            | –                    | –               | –             |
> | **TRELLIS + PhysPre** | 24.31           | 13.2            | 76.9            | 13.21            | 8.63       | 7.23         | 0.24                 | 0.12            | 6.55          |
> | **PhysXGen**        | **24.53**       | **12.7**        | **77.3**        | **7.24**         | **13.01**  | **11.30**    | **0.33**             | **0.08**        | **10.11**     |
>
>
> > **Weakness 4**: Concerns regarding the novelty of the dataset.
>
> **Answer 4**: Thank you for the constructive comment. While our dataset is based on PartNet, we have manually refined and integrated the original data to improve consistency and usability, and we have also added new physical attributes. Furthermore, we are actively expanding the dataset in two directions: (1) generating entirely new objects—**beyond those in PartNet**—using procedural generation methods, and (2) incorporating **more diverse 3D data** sources (Objaverse).
>
>
> > **Weakness 5**: Question about our merge processing.
>
> **Answer 5**: Thank you for pointing this out. We manually refine the results of the merging process to ensure that the merged outputs are reasonable and consistent. We will include an explanation of this step in the revised version of the paper.

---

> ### Author Response · Authors · 2025-08-03
>
> We have conducted additional experiments and provided new clarifications based on your suggestions. We sincerely hope these updates address your concerns, and we would truly appreciate it if you could review our responses.

---

### Official Review · Reviewer_zbQv · 2025-06-29

**Clarity:** 4
**Significance:** 3
**Originality:** 3
**Rating:** 5
**Confidence:** 4

**Summary:**

This paper introduces PhysXNet, a new dataset with rich part-level physical annotations, and a corresponding new physical generative model PhysXGen. PhysXNet is semi-automatically annotated with a human-in-the-loop review, ensuring its high quality. PhysGen further demonstrates the effectiveness of this dataset.

**Questions:**

- Are PhysXNet and PhysXPartNet the same thing?
- What is the metric for 'material'? I suggest authors add a brief introduction about metrics.

**Ethical Concerns:**

["NO or VERY MINOR ethics concerns only"]

**Final Justification:**

Since my concerns are addressed, I keep my original rating.

**Limitations:**

Yes

**Quality:**

4

**Strengths And Weaknesses:**

### Strengths
- A new dataset with rich physical annotations. Happy to see a new PartNet-like dataset built with the help of VLM. This will advance physical-based generation and related fields such as robotics.
- A new method based on the dataset introduced, showing better performance than the combination of previous methods.

### Minor Weaknesses
- The design of PhysXGen is not very novel. It appears to be a new branch of adaptor based on pre-trained models. I suggest that authors show the computational cost of the adaptor and the whole model. It would be great if physical parameters could be predicted efficiently.

---

> ### Author Rebuttal · Authors · 2025-07-29
>
> We thank you for your meticulous comments and suggestions to improve our work. We have carefully addressed all your comments one-by-one as follows:
>
> > **Weakness 1**: Missing computational cost of model.
>
> **Answer 1:** Thank you for the insightful comment. We have compared our model with the original Trellis model in terms of the number of parameters and inference time (evaluated on an NVIDIA A800 GPU). The results show that our model achieves impressive efficiency.
> | Methods | Parameters| Inference time|
> |-------|------|----|
> | Trellis | 1.2B  | 12.3s |
> | PhysXGen  | 1.87B | 15.4s|
>
> > **Weakness 2**: Naming inconsistency
>
> **Answer 2:** Yes, we apologize for the typo and will correct it in the revised version of the paper.
>
> > **Weakness 3**: Lack of a brief introduction about metrics.
>
> **Answer 3:** Thank you for the valuable suggestion. We have revised several evaluation metrics in the paper to better align with different physical aspects of the task. Additionally, we will include a brief introduction to these metrics to clarify them.

---

> > ### Comment · Reviewer_zbQv · 2025-08-03
> > **Acknowledgement**
> >
> > Thank the authors for the detailed feedback! My minor concerns have been addressed, and I will keep my original rating.

---

> > > ### Author Response · Authors · 2025-08-03
> > >
> > > We sincerely thank you for taking the time to review our work and provide valuable feedback.

---

### Official Review · Reviewer_aaVJ · 2025-07-02

**Clarity:** 3
**Significance:** 3
**Originality:** 4
**Rating:** 5
**Confidence:** 4

**Summary:**

The paper introduces PhysX, an end-to-end paradigm for generating 3D assets with physical properties, addressing a key gap in current 3D generative models. A central contribution is PhysXNet, "the first comprehensive physics-grounded 3D dataset,  comprising over 26K richly annotated 3D objects with fine-grained physical attributes like scale, material, and kinematics. This dataset, built via a "scalable human-in-the-loop annotation pipeline," enables PhysXGen, a novel framework that integrates physical knowledge into 3D asset generation, demonstrating strong performance in creating physically plausible models.

**Questions:**

1. What is the estimated accuracy of the VLM's initial annotations for each property type (e.g., material, affordance, kinematics), and what is the average human time/effort required for the "expert refinement" per object, especially for complex kinematic behaviors?

2. How well would PhysXGen generalize to generating physical properties for 3D objects belonging to entirely novel categories or exhibiting physical behaviors not extensively represented in the PhysXNet dataset?

**Ethical Concerns:**

["NO or VERY MINOR ethics concerns only"]

**Final Justification:**

Thank you for the detailed rebuttal. It has addressed all of my concerns. I will keep my original score.

**Limitations:**

Yes

**Quality:**

3

**Strengths And Weaknesses:**

Strengths

- PhysXNet is the first dataset to systematically annotate 3D objects with a wide range of physical properties (scale, material, affordance, kinematics, function descriptions) at a fine-grained, part-level granularity. This goes beyond existing datasets that often provide only object-level or limited physical information.
- With over 26,000 objects, PhysXNet provides a substantial foundation for training and evaluating physics-aware 3D models.
- The proposed human-in-the-loop pipeline, integrating vision-language models (GPT-40) for initial labeling and human refinement, is an efficient and robust approach to creating such a detailed dataset.
- The paper proposes an end-to-end paradigm for physical-grounded 3D asset generation, addressing a critical and underexplored area in 3D content creation. This work has high potential impact for fields like robotics, simulation, and embodied AI.

Weakness

- The paper's comparisons primarily focus on geometry-centric 3D generation or separate physical predictors. It would strengthen the evaluation to include comparisons with methods that directly learn material concepts from images/videos, such as those proposed by Zhai et al. (2024) or Zhong et al. (2024), to better contextualize PhysXGen's performance in specific physical property prediction.

- While the paper demonstrates performance on PhysXNet's test set, the generalization capability of PhysXGen to generating physical properties for 3D objects belonging to entirely novel categories or exhibiting physical behaviors (e.g., highly deformable objects, fluid interactions) not extensively represented in the PhysXNet dataset remains underexplored.

---

> ### Author Rebuttal · Authors · 2025-07-29
>
> We thank you for your valuable suggestions and efforts to refine our work. We have carefully addressed all your comments one-by-one as follows:
>
> > **Weakness 1**: Limited comparisons with approaches that directly learn properties from images/videos.
>
> **Answer 1**: We appreciate you for the careful suggestion. As detailed in our supplementary material, we have included the GPT-based predictor that learn from image during evaluations, where our method demonstrates impressive performance across most physical properties. Additionally, we provide a new comparison with OOAL (learn from image) [1] on affordance prediction, which further highlights the impressive performance of our approach.
>
> | Methods | Affordance|
> |-------|------|
> | OOAL [1] | 0.441  |
> | PhysXGen  | 0.372 |
>
> We would like to kindly clarify that the primary focus of our method is on capturing physically-grounded properties, rather than pursuing geometric fidelity alone. Furthermore, it is important to note that our model achieves improvements while being trained on less data (PhysXNet), compared to original Trellis, which relies on the much larger Objaverse dataset.
>
> [1] Li G, Sun D, Sevilla-Lara L, et al. One-shot open affordance learning with foundation models[C]//Proceedings of the IEEE/CVF Conference on Computer Vision and Pattern Recognition. 2024: 3086-3096.
>
> > **Weakness 2**: Limited evaluations on novel cases and novel categories.
>
> **Answer 2**: We thank the reviewer for the helpful suggestion. To further evaluate the generalization ability of our method, we have conducted an additional user study. In this study, we assess our method's performance on novel objects and novel categories, particularly focusing on deformable objects and selected physical attributes. Participants were asked to rate the plausibility of predicted properties on a 5-point scale: 5 – Highly Reasonable, 4 – Mostly Reasonable, 3 – Somewhat Reasonable, 2 – Barely Reasonable, 1 – Completely Unreasonable.
>
> After collecting the scores, we normalized them and present the results in the table below for clarity. It prove the promising generalization capability of PhysXGen in some physical properties.
>
> |          |                         |                         |                         |                         |
> | :------- | :---------------------- | :---------------------- | :---------------------- | :---------------------- |
> |          | **Novel cases**          |                         | **Novel categories**      |                         |
> | **Methods** | **Affordance**         | **Material**           | **Physical scale**      | **Material**            |
> | **Baseline** | 0.515                  | 0.547                  | 0.523                 | 0.481                  |
> | **PhysXGen** | **0.811**              | **0.780**              | **0.760**              | **0.742**              |
>
> To fundamentally address this generalization challenge, we are actively extending our dataset using procedural generation and annotating more diverse 3D data.
>
> > **Weakness 3**: Detailed questions about our annotation pipeline
>
> **Answer 3**: We appreciate you for this detailed question. The initial annotation quality from VLM models is closely related to the complexity of the physical property. For relatively straightforward properties such as physical dimensions, material (density), and functional descriptions, the success rate exceeds 90%. For more challenging attributes like affordance, the success rate drops to around 60–70%. For the most difficult property—kinematic information—the total success rate is close to zero. However, it can provide some useful information such as kinematic group information and neighboring parts. Based on it, kinematic-related annotations are manually annoated, which takes approximately 1 minute per object on average.

---

> ### Author Response · Authors · 2025-08-03
>
> We sincerely hope that our responses can help clarify your concerns. We would be very grateful if you could take a moment to review our rebuttal.

---

> > ### Comment · Reviewer_aaVJ · 2025-08-04
> >
> > Thank you for the detailed rebuttal. It has addressed all of my concerns. I will keep my original score.

---

### Official Review · Reviewer_NyNc · 2025-07-03

**Clarity:** 3
**Significance:** 4
**Originality:** 3
**Rating:** 5
**Confidence:** 4

**Summary:**

This paper introduces PhysX, an end-to-end pipeline for physical-grounded 3D asset generation. The core contributions are the creation of PhysXNet, a large-scale, systematically annotated 3D dataset featuring part-level physical properties (such as absolute scale, material, affordance, kinematics, and rich function descriptions), and PhysXGen, a feed-forward dual-branch generative model that jointly models geometry and physical attributes for 3D asset synthesis. PhysXNet uses a human-in-the-loop annotation pipeline leveraging vision-language models for scalable property extraction, and PhysXGen combines geometry- and physics-aware components to inject physical realism in 3D synthesis. The paper demonstrates advantages over baselines through quantitative metrics and qualitative visualization, and benchmarks the impact of architectural choices via ablation studies.

**Questions:**

1: Where is the original data (such as paired part meshes and materials) from? Is it only from PartNet?

2: Do authors try to curate existing datasets into PhysGen to scale up the dataset, although some of them may miss some kinds of annotations?

**Ethical Concerns:**

["Major Concern: Data privacy, copyright, and consent"]

**Final Justification:**

This is a solid paper with comprehensive experiments and analysis. I maintain my rating of accept.

**Limitations:**

Yes.

**Paper Formatting Concerns:**

N/A.

**Quality:**

4

**Strengths And Weaknesses:**

Strengths:

1: The introduction of PhysXNet is a strong contribution, as summarized in Table 1 and discussed in Section 3. It offers over 26K 3D objects annotated with five key physical properties at the part level, which fills a significant gap in current 3D datasets. Its human-in-the-loop annotation pipeline (Fig. 2) with VLM assistance is thoughtfully developed and demonstrated to be practical and scalable.

2: The taxonomy of physical properties (absolute scale, material and density, affordance, kinematics, and function description) is systematically discussed and collected, which is important for both simulation and embodied AI robotics tasks.

3: The paper supports its claims with thorough empirical results. Experimental results show that the proposed dataset can facilitate 3D generation methods.

4: The paper is well organized and written. Figures 4, 5, and 6 clearly show the architectural layout, generated outputs, and property-specific qualitative comparison, providing strong visual evidence of the method's efficacy. For example, Table 2 quantitatively demonstrates improvements in both physical property predictions and geometric fidelity, central to the paper's claim of joint modeling. Figure 6 visualizes similarity scores for physical attributes, supporting detailed comparative claims.

5: The PhysXGen model is built on a dual-branch architecture that effectively utilizes pretrained structural priors and jointly learns correlations with physical properties. The architecture choices are clearly described and reasoned. The use of VAEs and conditional diffusion models for both geometry and physics is up to date, and training protocols appear sound.

Weaknesses:

1: **The experimental baselines are somewhat limited.** Although TRELLIS and TRELLIS+PhysPre are sensible, they are little compared to other recent strong generative models, especially those capable of handling physical aspects or integrating physical priors. The performance improvement is a somewhat narrow scope of empirical benchmarking (PSNR and CD).

2: **Generalization and Diversity**. While the dataset is larger than some previous efforts, its generalization to other data domains (e.g., real-world scanned data, non-PartNet-derived assets) is not addressed. The claim of generalization is not fully substantiated with, for example, cross-dataset or out-of-distribution evaluations.

3: **The applications are limited.** The authors only discuss that the proposed dataset can be used to generate physical 3D objects. From my understanding, the proposed dataset can also help 3D part-level perception models (like 3D parts segmentation) and part-based 3D generation models. It would be promising if the dataset could also improve the performance on these tasks (or this question could be simply left to the research community).

4: **Physical Dimension Evaluation**. The evaluation of physical properties is reduced to a Euclidean comparison between rendered maps, which may not fully capture the semantics of errors in properties such as kinematics or affordances. This could be especially problematic for parts of the evaluation with a strong qualitative or functional component.

Although I raise some minor weaknesses, it is noted that this paper makes a strong contribution to collecting data, along with many necessary physical attributes. Therefore, I vote for acceptance at this time and hope the authors could open-source the collected dataset.

---

> ### Author Rebuttal · Authors · 2025-07-29
>
> We sincerely thank you for the encouraging comments and we have carefully addressed all your comments one-by-one as follows:
>
> > **Weakness 1**:  Limited experimental baselines.
>
> **Answer 1**: We appreciate you for the careful suggestion. As detailed in our supplementary material, we have included the GPT-based predictor during evaluations, where our method demonstrates superior performance across most physical properties. Additionally, we provide a new comparison with OOAL [1] on affordance prediction, which further highlights the impressive performance of our approach.
> | Methods | Affordance|
> |-------|------|
> | OOAL [1] | 0.441  |
> | PhysXGen  | 0.372 |
>
> We would like to kindly clarify that the primary focus of our method is on capturing physically-grounded properties, rather than pursuing geometric fidelity alone. Furthermore, it is important to note that our model achieves improvements while being trained on less data (PhysXNet), compared to original Trellis, which relies on the much larger Objaverse dataset.
>
> [1] Li G, Sun D, Sevilla-Lara L, et al. One-shot open affordance learning with foundation models[C]//Proceedings of the IEEE/CVF Conference on Computer Vision and Pattern Recognition. 2024: 3086-3096.
>
> > **Weakness 2**:  Generalization and diversity problem & data sources
>
> **Answer 2**: Thank you for this advice. Currently, our dataset is primarily based on PartNet. However, we would like to emphasize the strong scalability of our framework. On one hand, our human-in-the-loop annotation pipeline can be readily extended to real-world scans and other 3D datasets beyond PartNet. On the other hand, we also leverage procedural generation to expand the dataset. In fact, we are actively working on incorporating more diverse data sources and extending more data using procedural generation.
>
> > **Weakness 3**:  Limited applications
>
> **Answer 3**: Thank you for the valuable suggestion. While our dataset is designed with a focus on physically-grounded 3D asset generation in this work, we believe it has broad applicability to a range of related tasks such as 3D affordance prediction, 3D segmentation, and part-level generation. The dataset offers rich physical annotations that could benefit these tasks by providing a deeper understanding of object functionality and interaction. We view this work as a foundational step, and we plan to explore and support these downstream applications in future work.
>
> > **Weakness 4**:  Unconventional metrics in physical dimension evaluations.
>
> **Answer 4**: We sincerely thank you for your insightful comments. In response, we have revised some evaluation metrics in the paper to better align with different physical aspects. Specifically, we now evaluate absolute scale using Euclidean distance, density and affordance images via PSNR, kinematics with instantiation distance, and functional description through PSNR on cosine similarity score maps. We believe these metrics provide a more comprehensive assessment of the physical properties.
>
> | Methods             | Geometry        |                 |                 | Absolute scale ↓ | Material ↑ | Affordance ↑ | Kinematic parameters |                 | Description ↑ |
> | :------------------ | :-------------  | :-------------  | :-------------  | :--------------- | :--------- | :----------- | :------------------- | :-------------- | :------------ |
> |                     | **PSNR ↑**      | **CD ↓**        | **F-Score ↑**   |                  |            |              | **COV ↑**            | **MMD ↓**       |               |
> | **TRELLIS**    | 24.31           | 13.2            | 76.9            | –                | –          | –            | –                    | –               | –             |
> | **TRELLIS + PhysPre** | 24.31           | 13.2            | 76.9            | 13.21            | 8.63       | 7.23         | 0.24                 | 0.12            | 6.55          |
> | **PhysXGen**        | **24.53**       | **12.7**        | **77.3**        | **7.24**         | **13.01**  | **11.30**    | **0.33**             | **0.08**        | **10.11**     |
>
> > **Weakness 5**:  Curate existing datasets into PhysGen
>
> **Answer 5**: Thank you for your meticulous suggestion. As physical property learning typically requires a multi-task loss to jointly optimize diverse attributes, it becomes challenging to train effectively when only partial annotations are available. To address this issue, we are actively extending our dataset using procedural generation and annotating more diverse 3D data.

---

> ### Author Response · Authors · 2025-08-03
>
> We greatly appreciate your thoughtful comments and have carefully addressed each of them in our rebuttal. We kindly hope you will have a chance to review our responses.

---

> ### Comment · Reviewer_NyNc · 2025-08-06
>
> Thanks to the authors' comprehensive reply. My concerns have been addressed. I will keep my current rating.

---

### Comment · Area_Chair_k3Ue · 2025-08-01
**Discussion kick-off**

Hi everyone,

Thanks for all your hard work on this paper.

The discussion period is now open, and I encourage a productive exchange to clarify the remaining open questions. Reviewers: please engage with the authors early in this discussion window. Please don't hesitate to ask for further clarifications. A robust discussion is needed to make an informed decision.

Looking forward to hearing your thoughts. Let's have a productive chat to get this sorted out!

---

### Decision · Program_Chairs · 2025-09-17

**Decision:**

Accept (spotlight)

**Comment:**

This paper introduces PhysX, a framework for generating 3D assets with physical groundings, including a new dataset (PhysXNet) and a generative model (PhysXGen). The review process resulted in a clear consensus for acceptance. The reviewers recognized the novelty and significance of the contributions, particularly the new dataset.

Summary
The paper makes two primary contributions to bridge the gap between virtual 3D asset generation and real-world physical plausibility. First, it introduces PhysXNet, a 3D dataset systematically annotated with a rich set of fine-grained physical properties, including absolute scale, material, affordance, kinematics, and function descriptions. The dataset is constructed using an efficient human-in-the-loop pipeline that leverages vision-language models. Second, it proposes PhysXGen, a novel dual-branch generative framework that is trained on PhysXNet to explicitly model the latent correlations between 3D structure and physical properties, thereby generating assets that are both geometrically and physically plausible.

Strengths
There is a consensus among reviewers regarding the paper's strengths:
- The introduction of PhysXNet is a major contribution. It addresses a clear and important need in the field for 3D data with rich physical annotations. The scale of the dataset (~26K objects) and the comprehensive taxonomy of physical properties make it a valuable resource for future research in robotics and simulation.
- Moving 3D generative models beyond purely geometric and textural fidelity to encompass physical realism is a next step for real-world applications.
- The proposed PhysXGen model is well-designed, with a dual-branch architecture that effectively learns the joint distribution of shape and physics. The paper is supported by thorough experiments and ablation studies that demonstrate clear improvements over relevant baselines.

Weaknesses
While the overall assessment was positive, the initial reviews raised some valid points for clarification:
- Initial reviews questioned the validation process for the new dataset's annotations, particularly for complex properties like kinematics.
- The initial metrics used for evaluating some physical properties were noted as unconventional and could be better aligned with prior work.
- Some reviewers suggested that the experimental comparisons could be strengthened by including a wider range of baselines.

Reasons for Recommendation
The work is well-executed, clearly written, and addresses a timely and important problem. Researchers from various communities (e.g., 3D vision, robotics, and generative models) would benefit from this work.   a spotlight (SAC recommend).